# Investigating the impact of dehydration and hydration on In-Vivo hip soft tissue biomechanics

Fatemeh Khorami[1,2], Yalda Foroutan[3], Carolyn J. Sparrey[1,2]*

**1** Mechatronic Systems Engineering, Simon Fraser University, Surrey, British Columbia, Canada,
**2** International Collaboration on Repair Discoveries (ICORD), Vancouver, British Columbia, Canada,
**3** Computing Science, Simon Fraser University, Burnaby, British Columbia, Canada

* csparrey@sfu.ca

## Abstract

Hip soft tissue biomechanics has a significant effect on hip loading and fracture risk during falls. Despite the high dehydration rate in older adults, athletes, and outdoor workers and its association with a higher risk of falls, the effect of dehydration on hip soft tissue biomechanics is unknown. Twenty participants (13 females and 7 males, aged 18–35) underwent indentation tests and ultrasound imaging over the greater trochanter in both hydrated and dehydrated states. We assessed the hydration levels using a urine color chart and measured the tissue thicknesses via ultrasound. Results showed a significant increase in peak force (from $12.67 \pm 9.09$ N to $15.46 \pm 9.23$ N, $p < 0.05$) under dehydration. We observed notable sex differences, with males exhibiting higher stiffness and energy absorption than females, despite variations in peak force. Fat thickness emerged as a critical predictor of biomechanical response, particularly in the dehydrated state. These findings underscore the importance of hydration in maintaining soft tissue integrity and reducing injury risks. Future work should explore chronic dehydration effects and include broader demographic variations to enhance fall prevention strategies and clinical practice. This research also highlights the need for targeted hydration management in at-risk populations.

## Introduction

The prevalence of dehydration among athletes and workers in climate-vulnerable industries (e.g., construction and agriculture) due to heat exposure and heat strain is high, significantly impacting soft tissue properties and injury risk [1–3]. This condition not only compromises overall health but is also linked with an increased risk of falls [4]. This association is attributed to the adverse effects dehydration can have on balance [5], coordination [6], and muscle strength [7]. The hip is a critical focus in fall injury studies due to its vulnerability to serious injuries like fractures upon impact during falls among older adults. Studies have demonstrated the pivotal role of soft tissues, including muscles [8], tendons [9], and adipose tissue [10], in dissipating and distributing

**Data availability statement:** Data collected and used in this study are available on Zenodo at https://doi.org/10.5281/zenodo.15793643.

**Funding:** This research was supported by a Discovery Grant from the Natural Science and Engineering Research Council of Canada (NSERC RGPIN-2018-06382) and (NSERC RGPAS – 2018 - 522659).

**Competing interests:** The authors have declared that no competing interests exist.

impact forces and absorbing energy during falls [11]. However, the effect of dehydration on the mechanical behavior of hip soft tissues, including alterations in tissue elasticity, viscoelasticity, and frictional properties, remains a significant gap in current research. Characterizing the effect of hydration on hip soft tissues will help us more accurately simulate fall mechanics, quantify injury risk and inform injury prevention strategies.

The biomechanics of hip soft tissues, including their viscoelastic properties, play a crucial role in how energy is absorbed and dissipated during a fall [12]. Studies have shown that greater soft tissue thickness, often associated with higher body mass index, is linked to a reduced risk of hip fractures [13], highlighting the protective cushioning effect of these tissues [14,15]. However, gaps remain in our understanding of the precise mechanisms and the extent of variability in tissue response under different physiological conditions or states of hydration. Water content plays an important role in the biomechanics of muscles and skeletal soft tissues. For example, the stress-strain characteristics and ultimate failure properties of the collagenous structures that comprise ligaments and tendons are significantly affected by decreased water content [16]. Specifically, when the tissue's hydration is compromised at 40% solute concentration, the process of stress relaxation can extend up to 232% longer than in adequately hydrated conditions, likely due to diminished water-based lubrication [17]. A decrease in water content within tendons led to a contraction of the collagen fibers, consequently intensifying the magnitude of experienced tensile strains [16]. Furthermore, a reduction in water content diminished soft tissue sensitivity to strain [18] and reduced stress relaxations [19]. For example, research examining the human iliotibial band under varying osmotic pressures revealed a clear link between reduced water content and an increase in mechanical parameters like elastic modulus, tensile strength, and peak force, illustrating how hydration levels directly impact tissue mechanics [20]. Hydration significantly influences skin elasticity, a factor essential in maintaining the structural integrity and mechanical functionality of hip tissues [21]. Combining these studies suggests that hydration could affect hip soft tissue mechanics by affecting muscle, fat, and skin stiffness and energy absorption capabilities; however, the combined structural effect of hydration on hip soft tissues has not been measured.

The current study serves as an exploratory investigation into how hydration levels impact the biomechanical properties of hip soft tissues. We assessed tissue composition and observed changes in tissue thickness between hydrated and dehydrated states. This research provides initial quantification of the effects of hydration on tissue mechanics, highlighting the potential need for increased attention to hydration status. Additionally, understanding the impact of hydration on hip soft tissue biomechanics could inform rehabilitation protocols, refine fall simulations and injury risk models, enhance athletic performance, and contribute to the development of real-time hydration monitoring technologies.

## Methods

### Participants

Twenty volunteers ($23.3 \pm 3.6 \, \text{kg/m}^2$ BMI), (13 female and 7 male), (18–35 years) participated in the study. Participants with any pre-existing hip-related injuries were

excluded. At the beginning of the session, participants were asked to report their height and weight, which were used to calculate their BMI. The BMI classification used in this study identified individuals as underweight (BMI < 18.5), normal weight (18.5 ≤ BMI < 25), overweight (25 ≤ BMI < 30), or obese (BMI ≥ 30). All participants self-reported good general health to endure short-term dehydration. Ethical approval for this study was granted by the Simon Fraser University Ethics Board.

### Study design and hydration level assessment

The study was conducted over two sequential days, the study investigated biomechanical characteristics under different hydration states—hydrated and dehydrated. To control for variables, participants maintained a consistent diet and activity level across both days, documented via the Cronometer: The Nutrition Tracker (Revelstoke, British Columbia, Canada) [22,23]. On dehydration day, participants engaged in their regular activities but refrained from fluid intake (from 12 am to 2 pm) to induce mild, acute dehydration. On hydration day, participants followed a similar routine with the addition of water intake, adhering to the recommended water intake guidelines by the European Food Safety Authority (EFSA) [24]. The recommended intake is 2 L/day for females and 2.5 L/day for males [24]. For our test, participants consumed nearly half of this recommended daily intake before 2 pm. The order of testing (hydrated vs. dehydrated) was balanced among participants and food intake was monitored to ensure consistency across test days [25]. Upon arrival at the laboratory each day, participants assessed their hydration level by visually inspecting their midstream urine and comparing it to an 8-colour urine chart, categorizing hydration levels as well hydrated (urine color 1–3), euhydrated (4–6), or hypohydrated (7–8) **Fig 1**. These categories facilitated the monitoring of participants' hydration levels and were chosen based on previous research, which substantiated urine color as a valid, non-invasive indicator for adult hydration assessment [26–29].

### Indentation testing and mathematical modeling

Indentation tests were conducted over the greater trochanter of each participant using a 3-mm indenter, with cyclic loading at 50 Hz and a peak-to-peak displacement of 10 mm after an initial precompression of 5 mm **Fig 2**. The indentation experimental protocol was previously described [30]. Briefly, participants were positioned side-lying with their hips and knees flexed at approximately 45°, to expose the lateral surface of the hip. The greater trochanter was manually palpated

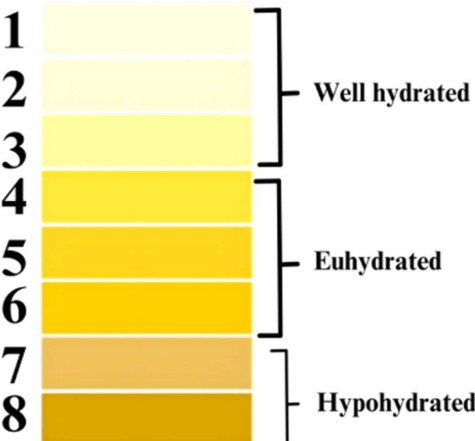

**Fig 1. Urine Hydration Assessment Chart.** This chart presents an 8-color urine hydration indicator ranging from pale straw to deep amber. Each color corresponds to a specific hydration level, with the lighter shades indicating optimal hydration and progressively darker shades signifying increasing levels of dehydration.

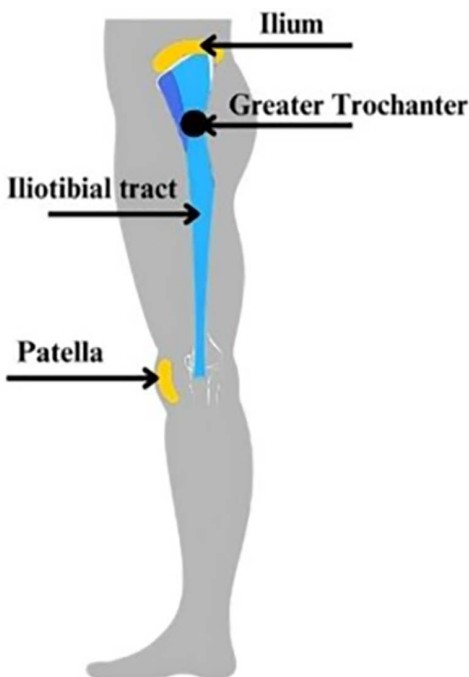

**Fig 2. The ilium and greater trochanter region, detailing where indentation tests were conducted.**

by the researcher. A semi-permanent marker was used to mark the location on the skin to ensure consistency between testing days and measurements. Ultrasound was used to verify that the indentation location was centered over the greater trochanter. Control limitations at high frequency limited the actual peak to peak displacement to 8 mm. Twenty cycles ensured consistent tissue response, and tests were repeated for reliability.

Force-displacement data was captured during the loading and unloading phases of each test cycle. For the loading phase, when the displacement velocity was non-negative, and similarly for the unloading phase when the displacement velocity was non-positive, we applied the following relationship:

$$
F = \begin{cases} F_{e-loading}\left[1 + c\dot{\delta}\right] & \text{for loading where } \dot{\delta} \geq 0 \\ F_{e-unloading}\left[1 + c\dot{\delta}\right] & \text{for unloading where } \quad \dot{\delta} \leq 0 \end{cases}
$$

In the application of this model, the force equation took the form of $F = Kd^n$, where F is the force, K signifies the stiffness coefficient derived from the elastic part of the force-displacement curve, d represents the displacement, and n denotes the nonlinearity factor in the force-displacement relationship. We determined the damping component c in reference to force F through the equation $c = c^* + \epsilon$, where a negative $\epsilon$ value indicated an underestimation of damping effects, while a positive value suggested an overestimation relative to the actual conditions.

## Ultrasound imaging and data analysis

To further understand the impact of hydration on soft tissue composition, we measured the thickness of skin, fat, and muscle over the greater trochanter area for each participant in both hydrated and dehydrated states. These measurements were taken using ultrasound imaging (L7 (4–13 MHz), Clarius Mobile Health) at the marked location used for the

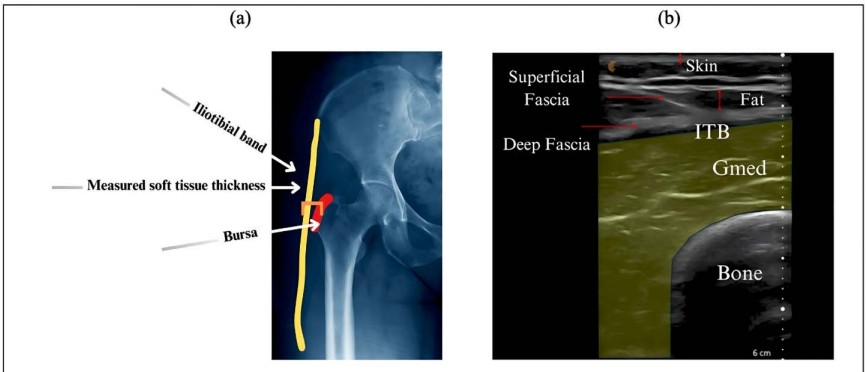

**Fig 3. a) The locations of soft tissue thickness measurements around the greater trochanter.** The yellow line represents the iliotibial band, the orange bracket indicates the measured soft tissue thickness, and the red line marks the Bursa. b) A sample ultrasound image showing the measured muscle, fat, and skin thickness.

indentation tests **Fig 3**. The participants' soft tissue composition was analyzed individually, and changes in the thickness of each tissue type were recorded. Specific attention was paid to the changes in fat, muscle, and skin thickness between the hydrated and dehydrated states. This analysis aimed to identify distinct patterns of soft tissue response to hydration changes and to correlate these changes with participants' sex or BMI. Force and displacement data were recorded, and stiffness was determined via the Hunt and Crossley model, suitable for rigid body dynamics modeling. Energy absorption was calculated using trapezoidal numerical integration of the area within the force/deformation curve during a loading/unloading cycle. Statistical analysis was conducted using R software, maintaining a 0.05 significance level. The Shapiro-Wilk test was applied to assess the normality of data distributions. Due to non-normal distributions, the Wilcoxon signed-rank test was used to compare stiffness, energy, and peak force between the hydrated and dehydrated states. Effect sizes were evaluated using Cohen's d to understand the practical significance of observed differences. A two-factor repeated measures ANOVA was performed to analyze the effects of hydration state (within-subject factor) and sex (between-subject factor) on stiffness, energy, and peak force. Significant effects were further examined using pairwise comparisons with Bonferroni correction. To assess the combined effect of hydration state and sex on multiple dependent variables, a MANOVA was conducted. For significant multivariate effects, univariate ANOVAs and post-hoc tests were performed. To identify significant predictors of the changes in peak force, stiffness, and energy absorption due to dehydration, a machine learning approach was employed using Random Forest Regressor models. The predictors included BMI, skin thickness, fat thickness, and muscle thickness from both hydrated and dehydrated states. Additionally, mixed-effects models were applied to account for repeated measures within participants, treating participants as random effects to handle within-subject variability. Multiple regression analysis was applied to model the relationship between each dependent variable (energy, peak force, stiffness) and multiple predictors (hydration state, BMI, sex, muscle, fat, and skin thickness).

## Results

Urine color analysis revealed a variability in dehydration levels, ranging from 1 to 6 on the urine color scale. The differences in self-assessed hydration were notable, with some participants exhibiting up to four colors change between their hydrated and dehydrated states. This indicates significant variations in hydration levels among individuals. Most of the participants included in the study exhibited some degree of dehydration, with 2 individuals maintaining the same urine color post-dehydration **Fig 4**. Analysis of the mechanical properties at the greater trochanter under hydrated and dehydrated conditions showed a significant increase in peak force. Specifically, peak force increased from $12.67 \pm 9.09\,N$ in the hydrated

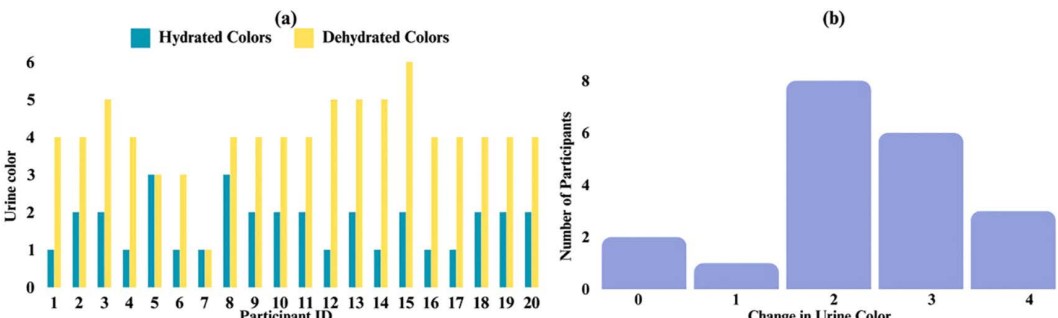

**Fig 4. (a) Urine color of 20 participants on dehydrated and hydrated test days, and (b) their differences, based on the 8-color urine chart.**

state to 15.46±9.23 N in the dehydrated state (W=49.0, p=0.036). The analysis revealed notable sex differences in the measured mechanical properties. Statistical analysis showed a significant difference in peak force between males and females (p=0.036). Specifically, in the hydrated state, males exhibited higher peak force values (mean of 13.50±9.10 N) compared to females, who had higher values in the dehydrated state (mean of 16.52±9.49 N) **Fig 5**. Additionally, MANOVA results indicated a significant effect of sex (p=0.03) on the combined dependent variables (energy, peak force, and stiffness). The mixed-effects model for peak force showed a significant effect of hydration state (p=0.02), with lower peak force observed in the hydrated state. Multiple regression analysis, including muscle, fat, and skin thickness, showed a significant effect of sex (p=0.02) on absorbed energy, with males showing higher absorbed energy levels. Furthermore, significant effects of both hydration state (p=0.04) and sex (p=0.01) were observed on peak force, with males exhibiting higher peak force and the hydrated state showing lower peak force. Soft tissue thickness over the greater trochanter varied significantly between hydrated and dehydrated states, and sex. Average tissue thickness decreased by ~7% from the hydrated state to the dehydrated state. Males had a mean hydrated tissue thickness ~14% less than females **Fig 6**.

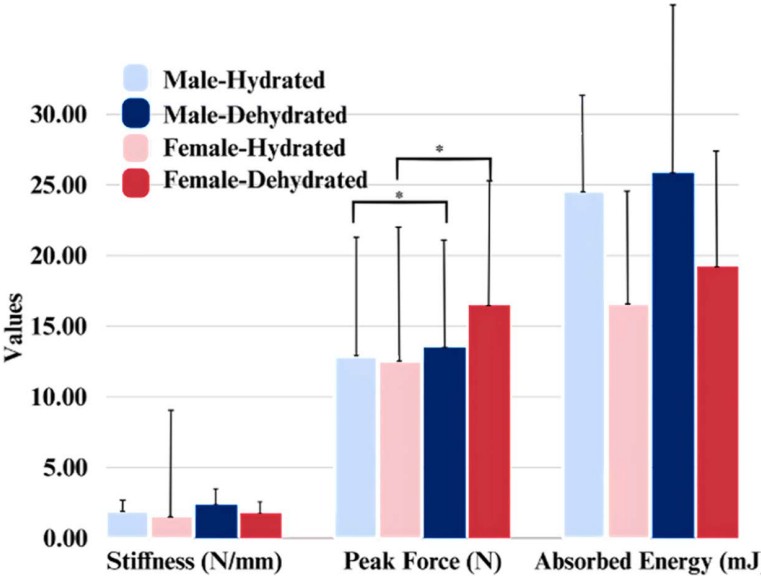

**Fig 5. Comparison of stiffness, absorbed energy, and peak force by sex and hydration state at the Greater Trochanter.** * Indicates significant differences.

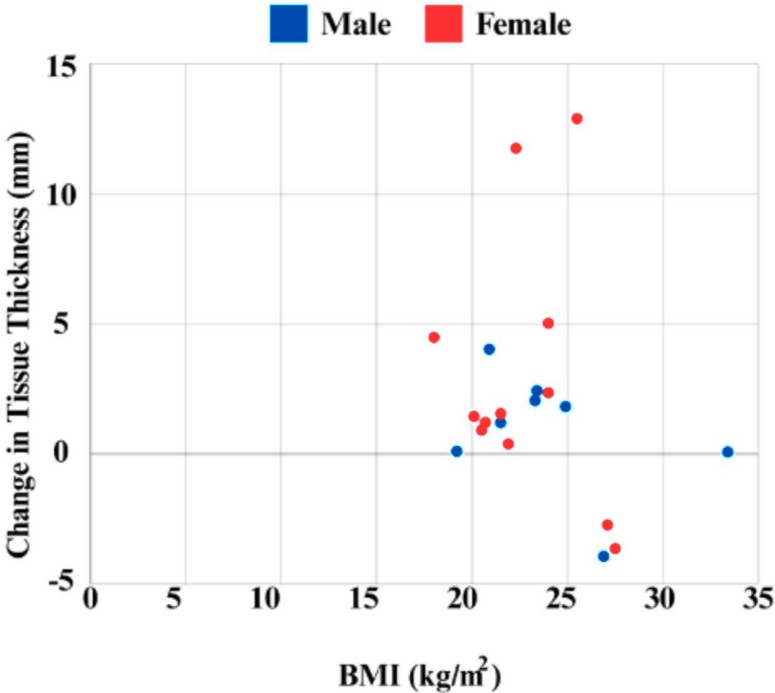

**Fig 6. BMI versus change in soft tissue thickness in hydrated to dehydrated state.** A positive value indicates a decrease in soft tissue thickness due to dehydration, while a negative value indicates an increase.

For the peak force difference, the Random Forest model demonstrated that the most important features for predicting peak force difference were fat thickness in the hydrated state (importance: 0.29), muscle thickness in the dehydrated state (importance: 0.20), and fat thickness in the dehydrated state (importance: 0.179). The most important feature for stiffness difference was fat thickness in the dehydrated state (importance: 0.51), followed by fat thickness in the hydrated state (importance: 0.14) and muscle thickness in the hydrated state (importance: 0.09). The most important features for predicting energy difference were fat thickness in the hydrated state (importance: 0.26), fat thickness in the dehydrated state (importance: 0.16), and muscle thickness in the dehydrated state (importance: 0.15). Our analysis revealed distinct patterns in how fat, muscle, and skin thicknesses responded to hydration changes among participants. Specifically, several participants (8 out of 20) showed an increase in fat thickness during the dehydrated state compared to the hydrated state. Notable examples include:

- A female participant with a BMI of 22.3 (urine color change from #2 to #4)

- A female participant with a BMI of 27.5 (urine color change from #2 to #4)

- A male participant with a BMI of 33.4 (urine color change from #2 to #5)

Conversely, most participants exhibited a decrease in muscle thickness in the dehydrated state. However, three individuals, all of whom were in the overweight category, showed an opposite trend with an increase in muscle thickness:

- A male participant with a BMI of 26.9 (urine color change from #1 to #4)

- A female participant with a BMI of 27.5 (urine color change from #2 to #4)

- A female participant with a BMI of 27.1 (urine color change from #2 to #4)

Additionally, our findings revealed a more consistent decrease in skin thickness across nearly all participants during dehydration. Only two participants exhibited a slight increase in skin thickness:

• A female participant with a BMI of 24 (urine color change from # 2 to #5)

• A female participant with a BMI of 24 (urine color change from # 2 to #4)

The overall decrease in skin thickness suggests that skin tissue may be particularly sensitive to fluid loss, impacting its elasticity and structural properties. The variability in these responses is shown in **Fig 7**.

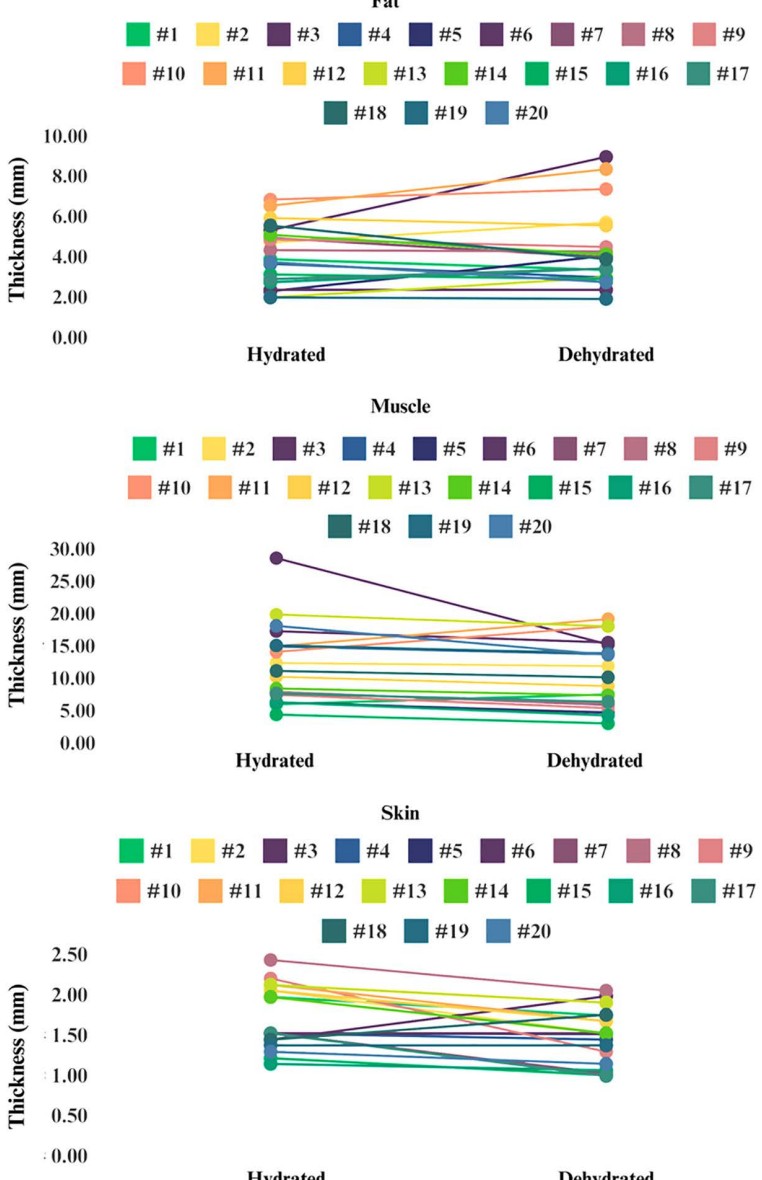

**Fig 7. The changes in fat (top panel), muscle (middle panel), and skin (bottom panel) thickness across 20 participants in hydrated and dehydrated states.** Each line represents an individual participant, highlighting the variability in responses to hydration changes.

## Discussion

The high prevalence of dehydration among athletes and outdoor workers (e.g., construction and agriculture) presents a significant public health challenge [31,32]. Dehydration can adversely affect balance [5], coordination [6], and muscle strength [33], thereby heightening the risk of falls and associated injuries [4]. By investigating the biomechanical properties of hip soft tissues under different hydration conditions, our study aims to uncover the biomechanical changes induced by dehydration on soft tissue over the greater trochanter.

### Strengths/limitations of the study

This exploratory study, while offering initial insights into the biomechanics of hip soft tissues in relation to hydration, is not without its limitations. One significant constraint was the reliance on self-reported measures for assessing hydration status. Quantitative indices of urine concentration, such as urine specific gravity (USG) and urine osmolality (Uosm), are reliable and objective markers of hydration status [34,35]. These methods provide precise quantification of hydration levels (using refractometry and osmometry respectively) and are commonly employed to detect subtle transitions between hydration and dehydration states [26]. While self-assessed urine color is inherently more subjective, validation studies have demonstrated strong correlations between urine color, USG, and Uosm [27,29], and evidence suggests that urine color assessment can approximate hydration status with reasonable accuracy for field and exploratory research applications. Therefore, given the exploratory nature of the present study, the use of self-reported urine color was considered an appropriate, validated, and minimally invasive method for monitoring relative hydration status. Another limitation of this study is its focus on acute dehydration, examining small changes over two sequential days. While this design offers preliminary insights on the effects of hydration on tissue mechanics, it does not capture the potential effects of chronic dehydration, which is more prevalent among older adults [36], particularly those in long-term care [37]. Chronic dehydration could affect soft tissue mechanics differently. This work provides important initial verification and proof of concept that hydration has a significant effect on soft tissue mechanics. This result justifies the increased complexity and impact of studying the effects of hydration on soft tissues in older adults in subsequent studies.

### Hydration's impact on soft tissue mechanics

Our investigation aligns with existing literature indicating that soft tissues, which are predominantly water, undergo significant mechanical property changes when water content varies. Majumder et al. (2008) reported that an 81% decrease in soft tissue thickness resulted in a 38% increase in peak force and a 97% increase in energy absorption [38]. While this study focused on the effects of tissue thickness rather than hydration, it emphasizes the importance of soft tissue thickness in biomechanical responses. In comparison, our study observed a 7% decrease in tissue thickness due to dehydration, leading to a 22% increase in peak force and approximately 11% increase in absorbed energy. Although the magnitude of thickness change in our study was less extreme, the observed increases in peak force and trend in absorbed energy reinforce the critical role of hydration in maintaining soft tissue mechanics. Soft tissues rely on water to maintain their compliance and viscoelastic properties. When tissues lose water, collagen fibers become less lubricated and more densely packed, increasing resistance to deformation [39]. Our data showed significant reductions in muscle thickness (from $32.45 \pm 7.34$ mm to $30.28 \pm 8.14$ mm, $p = 0.006$) and skin thickness (from $1.70 \pm 0.31$ mm to $1.64 \pm 0.32$ mm, $p = 0.019$) with dehydration. The tightening and reduced fluidity of the extracellular matrix likely contribute to the increased peak force observed [17]. Notably, even seemingly small changes in tissue properties, such as the approximately 20% increase in stiffness observed in our study, could have significant implications for the biomechanics of falls. Previous research has shown that variations in mechanical properties, such as tissue stiffness, can lead to substantial changes in impact forces [40]. For example, increasing joint stiffness by 25% has been found to result in up to a 150% increase in peak force during simulated falls [41]. These findings suggest that the 20% increase in tissue stiffness observed in our study could

correspond to a proportional increase in impact forces, potentially amplifying the risk and severity of injury during real-world falls. Importantly, our findings on the mechanical behavior of dehydrated live tissues may begin to mirror properties observed in cadaveric tissues. Specifically, post-mortem tissues exhibited up to a 100% increase in shear storage modulus [42,43]. This similarity raises critical questions about the extent to which dehydration affects the biomechanical properties observed in post-mortem studies, emphasizing the need to consider such factors when interpreting cadaveric data.

Our study observed significant sex differences in the mechanical properties of soft tissues, which we hypothesize may be attributed to inherent differences in tissue composition between males and females. Males generally have a higher muscle mass and denser connective tissue, contributing to the observed mechanical properties [44]. However, females typically have a higher fat content, which influences the distribution and retention of body water [45–47]. This higher fat content in females could explain the lower water content per kilogram of body weight at equivalent BMI levels, as fat tissue contains less water than muscle tissue [46]. It is important to note that BMI alone may not fully capture these differences because BMI does not account for the varying densities of muscle and fat tissues [48]. Moreover, the proportion of total body water (TBW) that is composed of extracellular water is typically higher in males than in females, particularly as BMI increases [46]. This difference is supported by studies showing that extracellular water expands with increased fat mass [49]. In our study, females exhibited greater soft tissue thickness in both hydrated ($34.05 \pm 6.88$ mm) and dehydrated ($31.30 \pm 8.64$ mm) states compared to males (hydrated: $29.48 \pm 7.75$ mm, dehydrated: $28.39 \pm 7.36$ mm). The observed lower peak forces in the hydrated state for females could be attributed to the higher water content in their tissues, which provides greater cushioning and reduces stiffness, leading to lower resistance to deformation. However, in the dehydrated state, the reduction in water content likely led to increased tissue stiffness and decreased cushioning, resulting in higher peak forces. This suggests that the changes in peak force with hydration status are more closely related to the hydration-dependent mechanical properties of the tissue, rather than just the tissue thickness.

Our findings reveal that the mechanical properties of soft tissues and their response to hydration levels may vary with BMI, particularly in overweight individuals. The distinct behavior of tissues in overweight subjects could reflect differences in body composition, namely the proportion of fatty tissue to lean mass, which influences water distribution and retention. The role of body composition becomes particularly relevant when considering the distribution of water and its biomechanical implications. Individuals with higher body mass index (BMI) typically have a greater proportion of adipose tissue, which contains less water than muscle tissue. This disparity likely explains the variations in mechanical properties observed across different BMI categories in our study. Specifically, individuals with higher fat content may exhibit a different response to dehydration, potentially due to less available water to lose from their adipose-rich tissues, affecting their overall tissue stiffness less dramatically. The variations in soft tissue thickness between different weight categories observed in our study could also be influenced by several other factors beyond body composition. For instance, the distribution of soft tissue in the body can be affected by an individual's metabolic rate, which in turn can impact the hydration levels within tissues [50]. Those with higher metabolic rates may process fluids differently, potentially leading to less water being retained in the soft tissues. Similarly, cardiovascular efficiency affects fluid distribution, with better circulation leading to more uniform hydration maintenance, contributing to the observed differences in soft tissue thickness. Additionally, the higher blood pressure often observed in obese individuals [51] may impact tissue mechanics by altering fluid distribution and vascular resistance [52]. This can affect the hydration dynamics within tissues, leading to less pronounced changes in stiffness and peak force upon dehydration. Moreover, obesity-related inflammation could degrade structural proteins like collagen, further influencing the mechanical response of tissues to hydration levels [53].

The mixed-effects models, which accounted for the repeated measures within participants, did not identify any statistically significant predictors among BMI, skin thickness, fat thickness, and muscle thickness for any of the outcome variables (the overall model performance was mixed, with the energy difference model showing the highest explanatory power ($R^2 = 0.300$). These results suggest that while fat and muscle thickness may have some influence on the biomechanical differences observed between hydrated and dehydrated states, fat thickness appears to have a greater effect

compared to muscle thickness. However, the overall model performance and the lack of significant predictors indicate that other factors might be influencing the outcomes, or that the sample size is insufficient to detect subtle effects.

### Clinical implications

The findings of this study indicate that dehydrated tissues exhibit increased peak forces for the same level of deformation. This suggests a heightened injury risk, particularly for individuals at risk of falls or recovering from hip injuries, as the increased forces are more likely to be transmitted to the bone, while the decreased tissue thickness leaves less soft tissue available for energy absorption. Given these insights, hydration management should be considered an important factor in clinical care, particularly for populations prone to dehydration, such as older adults, athletes, and workers in demanding environments. While the direct application of hydration assessments in addressing soft tissue biomechanical issues may require further research, the potential implications are clear: maintaining adequate hydration could play a critical role in reducing injury risk by preserving soft tissue properties that are essential for energy absorption during impacts. Long-term dehydration can lead to sustained cognitive impairments, which may contribute to an increased risk of falls and hip injuries, particularly in vulnerable populations [54,55]. Cognitive decline associated with chronic dehydration can impair balance, coordination, and decision-making [56]. In addition, novel imaging approaches for non-invasive, real-time hydration monitoring could be explored within the field of biomechanics [57,58].

### Conclusions

This study reveals that hydration significantly influenced the biomechanical properties of hip soft tissues. Importantly, these observations were also sex-specific with females showing a more significant effect of hydration on peak force. The observed increases in peak force and reduction in soft tissue thickness during dehydration underscore the importance of hydration in hip tissue mechanics. These initial insights motivate the need for further research to explore hydration's broader implications for musculoskeletal health and injury risk. As an exploratory study, these findings provide a foundational understanding that can inform the design of future research. Future studies should consider the inclusion of variables such as sex, body composition (including fat and muscle thickness), and baseline hydration status, as these factors appear to significantly influence the biomechanical response to hydration changes. Additionally, the impact of chronic versus acute dehydration should be further investigated to better understand the long-term implications for musculoskeletal health.

### Author contributions

**Conceptualization:** Carolyn J. Sparrey.

**Data curation:** Yalda Foroutan.

**Formal analysis:** Fatemeh Khorami.

**Funding acquisition:** Carolyn J. Sparrey.

**Investigation:** Carolyn J. Sparrey.

**Methodology:** Fatemeh Khorami.

**Project administration:** Carolyn J. Sparrey.

**Resources:** Carolyn J. Sparrey.

**Software:** Carolyn J. Sparrey.

**Supervision:** Fatemeh Khorami, Carolyn J. Sparrey.

**Writing – original draft:** Fatemeh Khorami.

**Writing – review & editing:** Yalda Foroutan, Carolyn J. Sparrey.

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
