## [Decision Letter · Decision Letter 0]

Dear Dr. Sparrey,

Thank you for submitting your manuscript to PLOS ONE. After careful consideration, we feel that it has merit but does not fully meet PLOS ONE’s publication criteria as it currently stands. Therefore, we invite you to submit a revised version of the manuscript that addresses the points raised during the review process.

We look forward to receiving your revised manuscript.

Kind regards,

Mario Milazzo

Academic Editor

PLOS ONE

Journal Requirements:

2 Thank you for stating in your Funding Statement:

“This research was supported by a Discovery Grant from the Natural Science and Engineering Research Council of Canada (NSERC RGPIN-2018-06382) and (NSERC RGPAS – 2018 - 522659)”

3. In the online submission form, you indicated that your data will be submitted to a repository upon acceptance.  We strongly recommend all authors deposit their data before acceptance, as the process can be lengthy and hold up publication timelines. Please note that, though access restrictions are acceptable now, your entire minimal  dataset will need to be made freely accessible if your manuscript is accepted for publication. This policy applies to all data except where public deposition would breach compliance with the protocol approved by your research ethics board. If you are unable to adhere to our open data policy, please kindly revise your statement to explain your reasoning and we will seek the editor's input on an exemption.

Additional Editor Comments (if provided):

After careful revision of the text, the manuscript can be re-evaluated after the authors address the points highlighted by the Reviewer.

Reviewers' comments:

Reviewer's Responses to Questions

**Comments to the Author**

1. Is the manuscript technically sound, and do the data support the conclusions?

Reviewer #1: Yes

2. Has the statistical analysis been performed appropriately and rigorously?

Reviewer #1: I Don't Know

3. Have the authors made all data underlying the findings in their manuscript fully available?

Reviewer #1: Yes

4. Is the manuscript presented in an intelligible fashion and written in standard English?

Reviewer #1: Yes

Reviewer #1: Summary:

In this study, the effects of acute dehydration on the mechanical properties of soft tissues were evaluated. To this end, 20 participants between ages 18 and 35 were recruited and underwent indentation imaging over the greater trochanter in both the hydrated and dehydrated states. Results showed that the peak indentation force increased under dehydration.

General Comments:

Please include line numbers to facilitate the review.

Detailed Comments:

Middle of second paragraph in Introduction: Please ensure that references have the same format throughout the manuscript. There is a non-numbered in-line reference at the end of the page whereas the rest of the references are numbered endnotes.

“Combined these studies suggest that hydration […]”: Please change the word “Combined” to “Combining”.

“The indentation tests were conducted over the greater trochanter using a 3-mm indenter [..]”: How was this location determined, and by whom? Was it done by palpation by the participant? I would imagine it might be more difficult to identify where the greater trochanter is on larger individuals.

Discussion section, Strengths/limitations of the study: Quantitative alternatives to the self-reported measures for assessing hydration status, such as urine specific gravity measurements, should be discussed.

**Do you want your identity to be public for this peer review?** For information about this choice, including consent withdrawal, please see our Privacy Policy

Reviewer #1: No

---

## [Author Response · Author response to Decision Letter 1]

23 Jun 2025

Dear Dr. Mario Milazzo,

Thank you for giving us the opportunity to submit a revised draft of our manuscript PONE-D-25-04801 entitled " Investigating the Impact of Dehydration and Hydration on In-Vivo Hip Soft Tissue Biomechanics" to the Journal of PLOS ONE. We appreciate the time and effort that you and the reviewers have dedicated to providing your valuable feedback on our manuscript. We are grateful to the reviewer for their insightful comments. We have been able to incorporate changes to reflect the suggestions provided by the reviewers. We have highlighted the changes within the manuscript. Here is a point-by-point response to the reviewers’ comments and concerns.

General Comments:

1.Please include line numbers to facilitate the review.

Apologies for the oversight on this. We have now added continuous line numbers throughout the manuscript. Note that references to line items in response to the reviewer comments are line items in the Manuscript file with track changes.

Detailed Comments:

2.Middle of second paragraph in Introduction: Please ensure that references have the same format throughout the manuscript. There is a non-numbered in-line reference at the end of the page whereas the rest of the references are numbered endnotes.

Thank you for pointing this out. We switched reference managers as we moved manuscript drafts between co-authors and lost track of a couple of references. We have corrected the formatting for these two papers and fixed additional citations in the wrong formatting on manuscript lines 253 and 274.

3.“Combined these studies suggest that hydration […]”: Please change the word “Combined” to “Combining”.

Thank you for catching this grammatical issue. We have revised the sentence to begin with “Combining”.

4.“The indentation tests were conducted over the greater trochanter using a 3-mm indenter [..]”: How was this location determined, and by whom? Was it done by palpation by the participant? I would imagine it might be more difficult to identify where the greater trochanter is on larger individuals.

Thank you for this question. In our study, the location over the greater trochanter was determined by the research team following a method similar to the one employed in our previous study -Ref 30.

Khorami F, Obaid N, Sparrey CJ. Sex differences in in vivo soft tissue compressive properties of the human hip in young adults: a comparison between passive vs active state. J Mech Behav Biomed Mater. 2025 May 1;165:106904.

We have added this methodological detail to the manuscript.

Page 5 line 114-119:

“The indentation experimental protocol was previously described (30). Briefly, participants were positioned side-lying with their hips and knees flexed at approximately 45°, to expose the lateral surface of the hip. The greater trochanter was manually palpated by the researcher. A semi-permanent marker was used to mark the location on the skin to ensure consistency between testing days and measurements. Ultrasound was used to verify that the indentation location was centered over the greater trochanter.”

and renamed the section as “Indentation testing and mathematical modeling” to better reflect the section content.

5.Discussion section, Strengths/limitations of the study: Quantitative alternatives to the self-reported measures for assessing hydration status, such as urine specific gravity measurements, should be discussed.

Thank you for this suggestion. We revised the strengths/limitations of the study section to discuss quantitative alternatives for assessing hydration status, specifically urine specific gravity (USG) and urine osmolality (Uosm). We highlighted that while these measures offer greater precision, they require sample collection and laboratory analysis, which increases invasiveness and logistical burden. We also noted that validation studies have demonstrated strong correlations between urine color and USG/Uosm, supporting the use of urine color as an appropriate, validated, and minimally invasive method for exploratory research. These points have been added to the revised manuscript, along with appropriate references.

Page 10, Lines 239-248:

“Quantitative indices of urine concentration, such as urine specific gravity (USG) and urine osmolality (Uosm), are reliable and objective markers of hydration status (34,35). These methods provide precise quantification of hydration levels (using refractometry and osmometry respectively) and are commonly employed to detect subtle transitions between hydration and dehydration states (26). While self-assessed urine color is inherently more subjective, validation studies have demonstrated strong correlations between urine color, USG, and Uosm (27,29), and evidence suggests that urine color assessment can approximate hydration status with reasonable accuracy for field and exploratory research applications. Therefore, given the exploratory nature of the present study, the use of self-reported urine color was considered an appropriate, validated, and minimally invasive method for monitoring relative hydration status.”

I have updated our funding statement in the cover letter as requested but also copy it here for completeness.

“This research was supported by a Discovery Grant and Accelerator Supplement from the Natural Science and Engineering Research Council of Canada (NSERC RGPIN-2018-06382 and NSERC RGPAS – 2018 – 522659 (CJS)). The funder had no role in study design, data collection and analysis, decision to publish, or preparation of the manuscript. There was no additional external funding received for this study.”

We believe that these revisions have significantly strengthened the manuscript. We hope that the changes made address the reviewers’ comments satisfactorily, and we look forward to the possibility of our revised manuscript being accepted for publication in PLOS ONE.

Sincerely

Dr. Carolyn Sparrey

---

## [Decision Letter · Decision Letter 1]

Investigating the Impact of Dehydration and Hydration on In-Vivo Hip Soft Tissue Biomechanics

PONE-D-25-04801R1

Dear Dr. Sparrey,

We’re pleased to inform you that your manuscript has been judged scientifically suitable for publication and will be formally accepted for publication once it meets all outstanding technical requirements.

Kind regards,

Mario Milazzo

Academic Editor

PLOS ONE

Reviewers' comments:

Reviewer's Responses to Questions

**Comments to the Author**

Reviewer #1: All comments have been addressed

2. Is the manuscript technically sound, and do the data support the conclusions?

Reviewer #1: Yes

3. Has the statistical analysis been performed appropriately and rigorously?

Reviewer #1: I Don't Know

4. Have the authors made all data underlying the findings in their manuscript fully available?

Reviewer #1: Yes

5. Is the manuscript presented in an intelligible fashion and written in standard English?

Reviewer #1: Yes

Reviewer #1: (No Response)

**Do you want your identity to be public for this peer review?** For information about this choice, including consent withdrawal, please see our Privacy Policy

Reviewer #1: No

---

## [Editor Report · Acceptance letter]

PONE-D-25-04801R1

PLOS ONE

Dear Dr. Sparrey,

I'm pleased to inform you that your manuscript has been deemed suitable for publication in PLOS ONE. Congratulations! Your manuscript is now being handed over to our production team.

Kind regards,

on behalf of

Dr. Mario Milazzo

Academic Editor

PLOS ONE